# The Activation of PPARγ by (2Z,4E,6E)-2-methoxyocta-2,4,6-trienoic Acid Counteracts the Epithelial–Mesenchymal Transition Process in Skin Carcinogenesis

**DOI:** 10.3390/cells12071007

**Published:** 2023-03-24

**Authors:** Enrica Flori, Sarah Mosca, Giorgia Cardinali, Stefania Briganti, Monica Ottaviani, Daniela Kovacs, Isabella Manni, Mauro Truglio, Arianna Mastrofrancesco, Marco Zaccarini, Carlo Cota, Giulia Piaggio, Mauro Picardo

**Affiliations:** 1Laboratory of Cutaneous Physiopathology and Integrated Center of Metabolomics Research, San Gallicano Dermatological Institute, IRCCS, 00144 Rome, Italy; 2SAFU Unit, Department of Research, Diagnosis and Innovative Technologies, IRCCS Regina Elena National Cancer Institute, 00144 Roma, Italy; 3Genetic Research, Molecular Biology and Dermatopathology Unit, San Gallicano Dermatological Institute, IRCCS, 00144 Rome, Italy; 4Faculty of Medicine, Unicamillus International Medical University, 00131 Rome, Italy

**Keywords:** skin carcinoma, PPARγ, EMT, octatrienoic acid

## Abstract

Cutaneous squamous cell carcinoma (cSCC) is the most common UV-induced keratinocyte-derived cancer, and its progression is characterized by the epithelial–mesenchymal transition (EMT) process. We previously demonstrated that PPARγ activation by 2,4,6-octatrienoic acid (Octa) prevents cutaneous UV damage. We investigated the possible role of the PPARγ activators Octa and the new compound (2Z,4E,6E)-2-methoxyocta-2,4,6-trienoic acid (A02) in targeting keratinocyte-derived skin cancer. Like Octa, A02 exerted a protective effect against UVB-induced oxidative stress and DNA damage in NHKs. In the squamous cell carcinoma A431 cells, A02 inhibited cell proliferation and increased differentiation markers’ expression. Moreover, Octa and even more A02 counteracted the TGF-β1-dependent increase in mesenchymal markers, intracellular ROS, the activation of EMT-related signal transduction pathways, and cells’ migratory capacity. Both compounds, especially A02, counterbalanced the TGF-β1-induced cell membrane lipid remodeling and the release of bioactive lipids involved in EMT. In vivo experiments on a murine model useful to study cell proliferation in adult animals showed the reduction of areas characterized by active cell proliferation in response to A02 topical treatment. In conclusion, targeting PPARγ may be useful for the prevention and treatment of keratinocyte-derived skin cancer.

## 1. Introduction

Cutaneous squamous cell carcinoma (cSCC) is the most common keratinocyte-derived cancer with metastatic potential, and it is associated with poor prognosis in the advanced stage [1,2]. Ultraviolet radiation (UVR) is the main causative agent involved in the development of skin cancer through multiple mechanisms, including induction of inflammation, oxidative stress, immunosuppression, DNA damage, and dysregulated signal transduction [3,4].

The peroxisome proliferator-activated receptors (PPARs) are nuclear receptor proteins that act as ligand-activated transcription factors to regulate various physiological processes, such as development, cell proliferation and differentiation, metabolism, and inflammatory responses. The three PPAR isoforms, PPARα, β/δ, and γ, exhibit different selectivity and responsiveness to ligands, transcriptional targets, and tissue distribution, although all three isotypes are expressed in human and mouse epidermis [5,6,7]. PPARγ was initially known for its ability to transactivate genes related to lipid metabolism, adipogenesis, and energy homeostasis [8,9,10]. In the skin, it has been shown to regulate a network of genes involved in cell proliferation, differentiation, and pigmentation, as well as inflammatory and antioxidant responses [7,11,12,13,14,15,16,17]. Moreover, PPARγ appears to play an important functional role in the control of skin barrier permeability, as an inhibitor of keratinocyte cell proliferation and promoter of epidermal terminal differentiation [5,18,19,20,21].

The progression of the skin carcinogenesis process is characterized by the occurrence of the epithelial–mesenchymal transition (EMT), a complex biological mechanism in which epithelial tumor cells tend to lose their differentiated properties (loss of E-cadherin, desmoplakin, and laminin-1 expression) and redirect to a mesenchymal-like phenotype (increased expression of N-cadherin, vimentin, fibronectin, MMP-2), with a consequent increase in their migratory and invasive potential [22,23,24,25,26,27]. Increasing evidence demonstrates that metabolic reprogramming is a hallmark of cancer and extensive metabolic dysregulation of cancer cells is related to the EMT program [28,29]. The Warburg effect, facilitating the production of energy from glycolysis and less from oxidative phosphorylation, is the most thoroughly described metabolic change able to potentiate the aggressive proliferation of cancer cells [30]. Moreover, tumor cells show dysregulated lipid metabolism, including high lipogenic and lipolytic capacity, elevated membranous lipid synthesis, and upregulation of bioactive lipid production, which induces EMT processes [31,32]. Based on the studies on lipid metabolism in cancer, the most extensive changes in lipid metabolism pathways are fatty acid (FA) metabolism, cholesterol metabolism, arachidonic acid metabolism, and PPAR signal transduction [33]. Increased expression and activity of stearoyl CoA desaturase (SCD1), i.e., the enzyme converting saturated fatty acids to Δ9-monounsaturated fatty acids, is involved in increased cancer cell proliferation, growth, migration, EMT, metastasis, and chemoresistance and maintenance of cancer stem cells [34]. An increase in cyclooxygenase-2 (COX-2) expression and a subsequent increase in intracellular prostaglandin E2 (PGE2) levels have been reported in malignant tumor cells, and many previous studies have indicated that these factors have roles in EMT induction [35,36].

Among the PPARs, PPARγ activation has been demonstrated to induce terminal differentiation and inhibit cell growth and inflammation in numerous malignant cell types and murine tumor models, suggesting that PPARγ agonists may act as tumor suppressors [16,18,20,37,38,39,40,41]. Furthermore, PPARγ activation promotes the reversion of the EMT process in different types of tumor cells. In lung adenocarcinoma cells treated with transforming growth factor β (TGF-β), a potent EMT inducer [42], PPARγ activation counteracts the loss of E-cadherin expression and inhibits the induction of mesenchymal markers, thus preventing migration and invasion [43]. The PPARγ agonist pioglitazone significantly influences EMT gene expression, promoting a more epithelial, and less mesenchymal, phenotype in a murine lung squamous cell carcinoma model [44]. Moreover, a new PPAR-γ ligand inhibits TGF-β1-induced EMT, migration, and invasion of papillary thyroid carcinoma cells through the p38 MAPK signaling pathway in a PPARγ-dependent manner [45]. Again, PPAR-γ activation has been shown to suppress the migration and invasion of triple-negative breast cancer cells through inhibition of the EMT process [46].

In previous studies, we demonstrated that the activation of PPARγ by the parrodiene derivative 2,4,6-octatrienoic acid (Octa) prevents cutaneous UV damage by acting on the different skin cell populations. In particular, Octa induces melanogenesis and antioxidant defense in human melanocytes and organ-cultured human skin [12] and counteracts senescence-like phenotype in human fibroblasts [11]. Most interestingly, the activation of PPARγ by Octa exerts a protective effect against UVA- and UVB-induced damage on normal human keratinocytes (NHKs), the major target of UV radiation. Specifically, Octa promotes antioxidant defense, greater cell survival, and the effective removal of UV-induced DNA damage in NHKs and human epidermal skin equivalents [47]. In addition, Babino et al. proved the good tolerability and long-term efficacy of a medical device containing Octa and urea in the reduction of grade III actinic keratoses (AKs) [48], skin lesions resulting from chronic and excessive UV exposure and having a certain risk of becoming cancerous [49,50]. More recently, it has been demonstrated that the topical application of a sunscreen containing inorganic filters (50+ SPF) and 0.1% Octa protects from sunburn cell formation, reduces the number of apoptotic keratinocytes, and prevents the main molecular alterations caused by UV radiation [51].

The present study aimed at investigating the possible role of the PPARγ activators Octa and the new compound (2Z,4E,6E)-2-methoxyocta-2,4,6-trienoic acid (A02), designed based on an in silico docking model built around the available structure of the PPARγ ligand-binding domain complexed with different ligands, in targeting keratinocyte-derived skin cancer. We evaluated the anti-proliferative/pro-differentiative efficacy and the ability of the compounds to counteract the EMT process in the A431 squamous cell carcinoma cell line. The in vitro results demonstrated the capacity of the PPARγ modulators to revert the EMT process induced by TGF-β1, with A02 demonstrating superiority in several endpoints. Therefore, in vivo experiments were performed to challenge the effects of A02 on DMBA-induced carcinogenesis [52] in a mouse model useful to trace proliferation in live animals, the MITO-Luc transgenic mouse [53]. Results from these studies strongly support the hypothesis that targeting PPARγ may be useful for the prevention and treatment of keratinocyte-derived skin cancer.

## 2. Materials and Methods

### 2.1. Materials

M154 defined medium, human keratinocyte growth supplements (HKGSs), fetal bovine serum (FBS), L-glutamine, penicillin/streptomycin, trypsin/EDTA, and D-PBS were purchased from Invitrogen Technologies (Monza, Italy). DMEM basal medium was purchased from Euroclone (Milan, Italy). β-actin antibody (A5441) (1:10,000), GAPDH antibody (G9545) (1:5000), catalase antibody (C0979) (1:1000), propidium iodide solution, GW9662, and 2′,7′-dichlorofluorescein diacetate (DCFH_2_-DA) were from Sigma-Aldrich (Milan, Italy). Aurum Total RNA Mini kit, SYBR Green PCR Master Mix, Bradford reagent were from Bio-Rad (Milan, Italy). RevertAid First Strand cDNA synthesis kit was from Thermo Fisher Scientific (Monza, Italy). The antibodies for p21 Waf1/Cip1 (#2947) (1:1000), phospho-p44/42 MAPK (ERK1/2) (Thr202/Tyr204) (#4370) (1:2000), phospho-p38 MAP kinase (Thr180/Tyr182) (#4511) (1:1000), p38 MAPK (#9212) (1:1000), PPARγ (#2430) (1:1000), phospho-AKT (Ser473) (#4060) (1:1000), AKT (#2920) (1:1000), phospho-histone H2A.X (#9718) (1:1000), cyclin D1 (#2978) (1:1000), secondary anti-mouse IgG HRP-conjugated antibody (#7076) (1:3000), and anti-rabbit IgG HRP-conjugated antibody (#7074) (1:8000) were purchased from Cell Signaling (Danvers, MA, USA); anti-ERK2 (SC-1647) (1:1000), anti-fibronectin (SC-8422) (1:1000), and anti-filaggrin (SC-66192) (1:250) were from Santa Cruz Biotechnology (Santa Cruz, CA, USA); anti-p53 (M7001) (1:1000) and E-cadherin (M3612) (1:1000) were purchased from DakoCytomation (Glostrup, Denmark); the antibodies for cytokeratin 10 (ab76318) (1:2000), involucrin (ab53112) (1:500), loricrin (ab85679) (1:500), and vimentin (ab92547) (1:1000) were purchased from Abcam (Cambridge, UK). Amersham ECL Western Blotting Detection Reagent was from GE Healthcare (Buckinghamshire, UK). RIPA lysis buffer, broad-spectrum protease inhibitor cocktail, and broad-spectrum phosphatase inhibitor cocktail were from Boster Biological Technology Co. (Pleasanton, CA, USA). Recombinant human TGF-β1 was purchased from PeproTech (Cranbury, NJ, USA). RNASE A was from Biobasic Canada Inc. (Markham, ON, Canada). Non-specific siRNA (sc-37007) and PPARγ siRNA (sc-29455) were from Santa Cruz Biotechnology (Santa Cruz, CA, USA). The Amaxa human keratinocyte Nucleofector kit was from Lonza (Basel, Switzerland). Dual-Luciferase Reporter assay system was purchased from Promega Corporation (Madison, WI, USA).

### 2.2. Ethics Approval and Consent to Participate

The Institute’s Research Ethics Committee (IFO) approval was obtained to collect samples of human material for research (Prot CE/286/06, approved on 21 April 2006). The study was conducted according to the principles of the Declaration of Helsinki. Human samples (primary cell cultures) were collected from patients who had provided written informed consent.

### 2.3. New Chemical Ligand

The molecular docking webserver SwissDock [54] was used to predict the molecular interactions that may occur between the target receptor PPARγ and the SPPARM (2Z,4E,6E)-2-methoxyocta-2,4,6-trienoic acid (A02), kindly supplied by PPM Services SA-A Nogra Group Company. The results were analyzed with UCSF Chimera viewer [55], using the known ligand Octa as a reference. For PPARγ, a model ligated with Octa was not available, but in general, all the structures listed in the database were fairly consistent in terms of binding pocket location and ligand orientation. This allowed us to select one of these structures from H. sapiens (2HFP), remove the ligand, and obtain a reliable apo structure for the docking procedure. The conformation highlighted in yellow produced the best energy values and no clashes, and it occupied the same space in the binding pocket as the original ligand for the holo 2HFP structure. This conformation was used as a reference for A02 docking, using the same method and parameters. The final conformation of A02 is shown in Figure 1A. The binding affinity, as estimated by the binding free energy between the different ligands and the receptor, demonstrated that A02 and Octa are comparable. The ligand–protein binding energies for A02 and Octa were −6.5431 kcal/mol and −6.5805 kcal/mol, respectively. However, for A02, there is indeed a solution with a better delta G value (−6.7746 kcal/mol) than the one chosen: the ligand was bound in a completely different location in the binding pocket when compared to the best solution for Octa and the original ligand of 2HFP. This does not mean that the conformation is incorrect or not viable, but an alternative conformation compared to the chosen reference.

### 2.4. Cell Cultures

Normal human keratinocytes (NHKs) were isolated from the neonatal foreskin following a previously described procedure [56]. NHKs were maintained at 37 °C under 5% CO_2_ in the defined medium M154 with HKGSs, L-glutamine (2 mM), penicillin (100 u/mL), and streptomycin (100 μg/mL). NHKs were sub-cultivated once a week, and the experiments were carried out in cells between passages 2 and 4. The A431 squamous carcinoma cell line was purchased from ATCC (Manassas, VA, USA) and was maintained in basal medium DMEM, supplemented with 10% FBS, L-glutamine (2 mM), penicillin (100 u/mL), and streptomycin (100 μg/mL), in a humidified atmosphere containing 5% CO_2_ at 37 °C. Cell cultures were routinely tested for Mycoplasma infection. For each experiment, at least three different donors were used. Cells were plated and 24 h later were stimulated with chemicals in fresh medium, in accordance with the experimental design.

### 2.5. UVB Irradiation of NHKs

NHKs were incubated in a medium without phenol red and irradiated with UVB at the dose of 25 mJ/cm^2^. Control cells were treated identically but without UVB exposure. The Bio-Sun irradiation apparatus (Vilbert Lourmat, Marne-la-Vallée, France) was employed. The UVB lamps emit ultraviolet rays between 280 and 320 nm, with peak luminosity at 312 nm. UVB lamps do not have UVC emission. UVB was supplied by a closely spaced array of two UVB lamps which delivered uniform irradiation at a distance of 10 cm. Based on a programmable microprocessor, the Bio-Sun system constantly monitors UV light emission. The irradiation stops automatically when the received energy matches the programmed energy (range of measure: 0 to 9999 J/cm^2^).

### 2.6. RNA Extraction and Real-Time RT-PCR

Total RNA was isolated using the Aurum Total RNA Mini kit, according to the manufacturer’s instructions. Total RNA samples were stored at −80 °C until use. Following DNAse I treatment, cDNA was synthesized using a mix of oligo-dT and random primers and RevertAid First Strand cDNA synthesis kit according to the manufacturer’s instructions. Real-time RT-PCR was performed in a total volume of 10 μL with SYBR Green PCR Master Mix and 200 nM concentration of each primer. Sequences of all primers used are shown in Appendix A. Reactions were carried out in triplicates using a CFx96 Real-Time System (Bio-Rad, Hercules, CA, USA). Melt curve analysis was performed to confirm the specificity of the amplified products. Expression of mRNA (relative) was normalized to the expression of *GAPDH* mRNA by the change in the Δ cycle threshold (ΔCt) method and calculated based on 2^−ΔCt^.

### 2.7. Western Blot Analysis

Cells were lysed in RIPA lysis buffer supplemented with a protease/phosphatase inhibitor cocktail and then sonicated. Total cell lysates were clarified by centrifugation at 12.000 rpm for 10 min at 4 °C and then stored at −80 °C until analysis. Following spectrophotometric protein measurement, equal amounts of protein were resolved on acrylamide SDS-PAGE and transferred onto a nitrocellulose membrane (Amersham Biosciences, Milan, Italy). Protein transfer efficiency was checked with Ponceau S staining (Sigma-Aldrich, St. Louis, MO, USA). Membranes were first washed with water, blocked with EveryBlot Blocking Buffer (Bio-Rad Laboratories S.r.l., Milan, Italy) for 10 min at room temperature, and then treated overnight at 4 °C with primary antibodies (according to data sheet instructions). A secondary anti-mouse IgG HRP-conjugated antibody or anti-rabbit IgG HRP-conjugated antibody was used. Antibody complexes were visualized using enhanced chemiluminescence (ECL) substrate. A subsequent hybridization with anti-β-actin or anti-GAPDH was used as a loading control. Protein levels were quantified by measuring the optical densities of specific bands using the UVI-TEC Imaging System (Cambridge, UK). The control value was taken as 1-fold in each case.

### 2.8. Transient Transfection and Luciferase Assay

NHKs were transfected with pGL3-(Jwt)3TKLuc reporter construct [57] using the Amaxa human keratinocyte Nucleofector kit, according to the manufacturer’s instructions. A pTK-Renilla-expressing vector was also transfected as an internal control. After 24 h of transfection, cells were exposed to A02 (10–90 μM) for 24 h. After treatment, cells were harvested in 100 μL lysis buffer, and 20 μL of the extract was assayed for luciferase activity using Promega’s Dual-Luciferase, according to the manufacturer’s protocol. The luciferase activity was expressed as fold increase in the activity with respect to that obtained in non-stimulated cells.

### 2.9. MTT Assay

Cells treated with Octa (90 μM) or A02 (90 μM) for 72 h were then incubated with 3-(4,5-dimethyl-2-thiazolyl)-2,5-diphenyl-2H-tetrazolium bromide (MTT) (1 mg/mL) for 2 h at 37 °C and lysed in dimethyl sulfoxide (DMSO). The absorbance at 570 nm was measured using a DTX880 Multimode Detector spectrophotometer (Beckman Coulter S.r.l., Milano, Italy). The measurement was performed in triplicate for each sample.

### 2.10. Cell Number Analysis

Cells were treated with Octa (90 μM) or A02 (90 μM) for 72 h and then detached by trypsinization. Cell number was evaluated by cell counting using an Axiovert 40C phase-contrast microscope (Zeiss, Milan, Italy). None of the employed chemicals exhibited positivity in the trypan blue exclusion assay test.

### 2.11. Flow Cytometric Analysis of Cell Cycle

After the treatment, cells were detached by trypsinization and fixed with ice-cold 70% ethanol solution overnight at 4 °C for maximum resolution of cellular DNA. The samples were stained in PBS containing propidium iodide (50 µg/mL) and RNAse A (40 µg/mL). The DNA content of the cells was measured using a MACSQuant Analyzer (Miltenyi Biotec GmbH; Bergisch Gladbach, Germany). In total, 10^4^ events were acquired for each sample. The measurement was performed in duplicate for each sample. Flow cytometric histograms were analyzed by defining borders between pre-G_1_, G_1_, S, and G_2_ + M phase populations. MACSQuantify Software was used to analyze data.

### 2.12. Immunofluorescence Analysis

A431 cells were fixed either with 4% paraformaldehyde followed by 0.1% Triton X-100 to allow permeabilization or with cold methanol at −20 °C. Cells were then incubated with the following primary antibodies: anti-vimentin rabbit polyclonal antibody (1:400) (Abcam, Cambridge, UK), anti-fibronectin mouse monoclonal antibody (1:400) (Santa Cruz Biotechnology, Santa Cruz, CA, USA), and anti-E-cadherin monoclonal antibody (1:200) (Dako, Santa Clara, CA, USA). The primary antibodies were visualized using goat anti-rabbit Alexa Fluor 546 conjugate, goat anti-mouse Alexa Fluor 546, and goat anti-mouse Alexa Fluor 488 conjugate antibodies (1:800) (ThermoFisher Scientific, Monza, Italy). Coverslips were mounted using ProLong Gold antifade reagent with DAPI (Invitrogen, Waltham, MA, USA). The fluorescence signal was evaluated by recording stained images, using a CCD camera (Zeiss, Oberkochen, Germany). For vimentin and fibronectin stainings, the number of positive cells was measured by counting at least 150 cells randomly taken from 10 different microscopic fields, and results are expressed as fold change with respect to control. Quantitative analysis of E-cadherin fluorescence intensity was performed using the Zen 2.6 (blue edition) software (Zeiss, Oberkochen, Germany) and the results are expressed as mean fluorescence intensity/cell ± SD.

### 2.13. Measurement of Intracellular Production of Reactive Oxygen Species (ROS)

Intracellular ROS production was determined with the fluorescent probe DCFH2-DA. At the end of the treatment, cells were incubated with 10 μM DCFH2-DA in D-MEM without phenol red for 30 min at 37 °C, washed with PBS, trypsinized, and centrifuged at 200× *g*; then, they were collected in 1 mL of PBS. ROS signals were measured by flow cytometry, using the MACSQuant Analyzer. In total, 10^4^ events were acquired for each sample. The measurement was performed in duplicate for each sample. MACSQuantify Software was used to analyze data. The intracellular ROS levels were quantified using the median of the FL1 channel of fluorescence because it matched the maximal number of cells with the highest fluorescence.

### 2.14. Scratch Assay

A431 cells were seeded on 35 mm Petri dishes and allowed to grow until confluence. A cell-free area was then generated by wounding the cell monolayer using a pipette tip. After repeated washes, cells were immediately fixed (time zero, T0) or treated with the two compounds in the presence or absence of TGF-β1 as described above and then fixed after 24 h. Images were recorded using a CCD camera, and the migratory ability was determined by measuring the edge distance using the Zen 2.6 software. The results are expressed as fold change with respect to T0 which was set as 1 by definition.

### 2.15. RNA Interference Experiments

For the RNA interference experiments, A431 cells were transfected with 100 pmol siRNA specific for PPARγ (sc-29455; Santa Cruz Biotechnology, Santa Cruz, CA, USA). An equivalent amount of non-specific siRNA (sc-37007; Santa Cruz Biotechnology, Santa Cruz, CA, USA) was used as a negative control. Cells were transfected using Amaxa human keratinocyte Nucleofector kit (Lonza, Basel, Switzerland), according to the manufacturer’s instructions. To ensure identical siRNA efficiency among the plates, cells were transfected together in a single cuvette and plated immediately after nucleofection. Twenty-four hours after transfection, treatments were added to some samples in agreement with the experimental design.

### 2.16. Lipid Extraction from A431 Cells

Lipids were extracted from A431 cells in accordance with the Bligh and Dryer procedure with slight modifications [58]. Briefly, lipids were extracted with chloroform:methanol (2:1) (2 × 1 mL) after the addition of butylhydroxytoluene to prevent oxidation of oxygen-sensitive compounds. Then, 75 µL of a mixture of deuterated standards (d6CH 4 µM; d17C16:0 8 µM; d98TG48:0 2.4 µM; d4PGD2 1 µM; d4PGE2 1 µM; d85HETE 1 µM) was added to control the analytical performance and to calculate the relative abundance of the lipid species detected. The organic layers were collected and evaporated under nitrogen. The dried lipid extract was dissolved in 75 µL of acetonitrile and stored at −80 °C until the analysis.

### 2.17. GC-MS Analysis of FAMEs

Bound fatty acids (FAs) were analyzed as FA methyl esters (FAMEs) obtained after the derivatization reaction described below. For the simultaneous saponification and methylation of bound FAs, 250 μL KOH solution (0.5 M) in anhydrous methanol was added to the dried extract and incubated at 37 °C for 20 min under constant shaking. Then, 0.5 mL HCl (0.25 M) was added to neutralize the alkaline reaction mixture. After vortex mixing, 0.25 mL K2SO4 (6.7%) and 1 mL hexane:isopropanol (3:2 *v*/*v*) containing 0.0025% BHT were added and vortexed. After centrifugation, the lipid-enriched upper phase was transferred to an Eppendorf tube and evaporated under nitrogen. The dried FAME extract was dissolved in 20 µL isopropanol (IPA) and was analyzed by gas chromatography–mass spectrometry (GC-MS) to establish FA profiles in the lipid extracts (GC 7890A coupled to MS 5975 VL analyzer, Agilent Technologies, Santa Clara, CA, USA). The chromatographic separation was carried out on an HP-FFAP (crosslinked FFAP, Agilent Technologies, Santa Clara, CA, USA) capillary column (length 50 m, film thickness 0.52 µm). Helium was used as the carrier gas. The initial GC oven temperature was 40 °C, and the GC oven temperature was linearly ramped up to 240 °C at 8 °C/min. The total run time was 60 min. The injector and the GC-MS transfer lines were kept at 230 °C and 250 °C, respectively. Total ion chromatograms (TICs) were acquired, and areas of single peaks, corresponding to the FAMEs, were integrated with the qualitative analysis software. The identity of the detected FAMEs was verified by comparison with authentic standards and matched with library spectral data. The pmol amounts of FAMEs were calculated against the pmol amounts of d31C16:0 generated from d98TG48:0.

### 2.18. GC-MS Lipid Analysis of FFAs and CH

Free fatty acids (FFAs) and cholesterol (CH) were analyzed after direct silylation with N,O-bis-(trimethylsilyl)-trifluoroacetamide containing 1% trimethylchlorosilane as catalyst (Sigma–Aldrich, Milan, Italy). After 30 min at 60 °C, the samples were analyzed with a GC 7890A coupled to an MS 5975 VL analyzer (Agilent Technologies, Santa Clara, CA, USA). The chromatographic separation was performed on an HP-5MS (Agilent Technologies, Santa Clara, CA, USA) capillary column (30 m × 250 µm × 0.25 μm), using helium as the carrier gas. An oven temperature gradient from 80 °C to 200 °C at 8 °C/min and then to 250 °C at 10 °C/min was used. The injector and the GC-MS transfer lines were kept at 260 °C and 280 °C, respectively. Samples were analyzed in scan mode utilizing electron impact (EI) mass spectrometry. The identity of the detected FFAs and CH was verified by the comparison with authentic standards and the matching with library spectral data. The area of the peaks corresponding to C16:0, C16:1, C18:0, C18:1, and CH was integrated with the qualitative analysis software. The pmol amount of FFAs and CH were calculated against the added pmol amounts of d6CH and d17C16:0, respectively.

### 2.19. Assessment of Pro-Inflammatory Lipid Mediators

Prostaglandins (PGD2, PGE2, PGF2α) and 5HETE release in cell media were measured by HPLC/MS/MS. A431 supernatants (1 mL) were combined with 10 μL of IS solution (PGD2-d4, PGE2-d4, 5HETE-d8 1 µM), 100 µL of 0.2% FAs, and 3 mL of ethyl acetate. Samples were vortexed and centrifuged at 20,000× *g* for 20 min at 4 °C. The extraction procedure was repeated twice, and the clear supernatants were collected and evaporated under N2. After evaporation, the samples were dissolved with 30 μL of acetonitrile containing BHT (1 mM) and stored at −80 °C before LC-MS/MS analysis. Chromatographic separation was carried out using the Agilent Technologies 1200 HPLC Liquid Chromatography System (Palo alto, CA, USA) with a C18 column (Symmetry, 3.5 μm, 100 mm × 2.1 mm, Waters) as previously reported [59,60]. Briefly, the mobile phase flow rate was 0.2 mL/min. Mobile phase A consisted of acetonitrile–water–formic acid (20:80:0.1, *v*/*v*/*v*), and mobile phase B consisted of acetonitrile–formic acid (100:0.1, *v*/*v*). Mobile phase B was increased from 0 to 100% in a linear gradient over 6 min and maintained at 100% until 10 min. Mobile phase B was then decreased to 0% from 10 to 11 min and maintained at 0% until 22 min. The column temperature was maintained at 40 °C. The injection volume was 10 μL. The overall run time was 25 min. Negative ion electrospray tandem mass spectrometry was carried out with an Agilent Technologies triple quadrupole 6400 mass spectrometer at unit resolution with multiple reaction monitoring (MRM) performed by monitoring the following transitions: PGD2 351 → 233; PGD2-d4 (IS for PGD2) 355 → 233; PGE2 351 → 315; PGE2-d4 (IS for PGE2) 355 → 319; PGF2α 353 → 193; PGF2α-d4 (IS for PGF2α) 357 → 197; 5HETE 319 → 115; 5HETE-d8 (IS for 5HETE) 327 → 116. For each compound to be quantified, an internal standard was selected and a linear curve was generated where the ratio of analyte standard peak area to internal standard peak area was plotted versus the amount of analyte standard. The results were calculated as nM/10^6^ cells and are reported as percent variation vs. control values.

### 2.20. In Vivo Study

Mice were cared for in accordance with the Principles of Laboratory Animal Care (National Institutes of Health Publication No. 85–23, revised 1985) and with national laws. The experimental protocols comply with the principles of the Declaration of Helsinki (https://www.nc3rs.org.uk/arrive-guidelines (accessed on 1 June 1964)) and were approved by the National Ethics Committee for Animal Experimentation of the Italian Ministry of Health. The mice were housed in single cages with wood-derived bedding material in a specific-pathogen-free facility with a 12 h light/dark cycle under controlled temperatures (20–22 °C). and received water and food ad libitum. All experimental procedures conformed to protocols approved by the Regina Elena National Cancer Institute Animal Care and Use Committee and were performed in accordance with the Guide for the Care and Use of Laboratory Animals and the guidelines of the National Institutes of Health, according to the current national legislation (Art. 31 D.lgs 26/2014, 4 March 2014). The animals used in the study were 6–8-week-old MITO-Luc mice [53] of both sexes, maintained on an FVB background.

### 2.21. Carcinogenesis Mouse Model and In Vivo Bioluminescence Imaging (BLI) Analysis

Skin papillomas were generated by local administration of 7,12-dimethylbenz(a)anthracene and the forbolic ester 12-O-tetradecanoylphorbol-13 acetate (DMBA-TPA) on the shaved ventral skin of MITO-Luc mice [61]. Mice were treated with a single topical application of 860 μM DMBA (Sigma-Aldrich, Milano, Italy) in 0.2 mL acetone or with the solvent alone. One week after, 100 μM of TPA (Alexis Biochemicals, San Diego, CA, USA) in 0.2 mL acetone or the solvent alone was topically applied twice weekly. After the appearance of papillomas, topical treatments with A02 (20 mg dissolved in 1 mL of DMSO and 4 mL glycerol) were performed five days weekly. Solvent alone was applied as a placebo.

In vivo bioluminescence BLI represents a new approach to molecular imaging. It allows us to visualize internally generated light linked to specific physiological and/or pathological cellular processes in small living animals [62]. In the MITO-Luc transgenic reporter mouse animal model, luciferase activity is under the control of the cyclin B2-dependent nuclear factor-Y (NF-Y) promoter and is consequently restricted to proliferating cells [63,64,65]. Bioluminescence imaging (BLI) analysis was performed using the IVIS Lumina II equipped with the Living Image 2.20 software package for data quantification (PerkinElmer, Waltham, MA, USA), [53]. For in vivo imaging, mice were anesthetized, and D-luciferin (75 mg/kg body weight) (PerkinElmer, Waltham, MA, USA) dissolved in phosphate-buffered saline (PBS) was administered i.p. Ten minutes later, quantification of light emission was performed in photons per second per square centimeter per steradian (photons/s/cm^2^/sr) and visualized in a pseudocolor scaling. Time exposure ranged from 1 to 5 min, depending on light intensity. Tumor area (cm^2^) was measured by automatically drawing the regions of interest upon BLI analysis using the Living Image 2.20 software.

### 2.22. Statistical Analysis

Data were represented as mean ± SD of three independent experiments using at least three different donors for primary cells. Statistical significance was assessed using paired Student’s *t*-test or ANOVA followed by Tukey’s multiple comparison test using GraphPad Prism (GraphPad Software, Boston, MA, USA). The minimal level of significance was *p* < 0.05.

## 3. Results

### 3.1. A02 Induces PPARγ Expression and Activation in NHKs

We initially verified the ability of A02 to activate PPARγ in keratinocytes (chemical structure and molecular docking of A02 shown in Figure 1A). For this purpose, we treated NHKs with different doses of A02 (10, 50, 90 μM) for 24 h, and we observed a significant induction of PPARγ protein expression starting from the dose of 50 μM (Figure 1B). A luciferase assay using a PPARγ reporter construct (pGL3-(Jwt)TKLuc) [57] consistently demonstrated a dose-dependent PPARγ activation upon 24 h A02 treatment (Figure 1C). These data indicate that A02 not only induces PPARγ protein expression but also leads to its activation.

### 3.2. Effects of A02 on Cellular Antioxidant Defense and DNA Damage in UVB-Irradiated NHKs

We previously demonstrated the ability of Octa to induce the antioxidant defense in NHKs [47]. Here we reproduced some experiments with A02 to evaluate its potential to exert a similar effect. The treatment with the compound (10, 50, 90 μM) for 6 h significantly increased the mRNA expression level of *CATALASE* (*CAT*), *HEME-OXIGENASE 1* (*HO-1)*, and *NAD(P)H QUINONE DEHYDROGENASE 1* (*NQO1)* (Figure 1D), three key enzymes involved in cellular defense against oxidative stress [66,67]. The Western blot analysis of catalase confirmed the gradual induction of the protein after 24 h of treatment (Figure 1E). Then we went on to investigate whether A02, similarly to Octa, may exert a protective effect against oxidative stress and DNA damage induced by UVB exposure in keratinocytes [47]. To this end, NHKs were pre-treated with A02 (10, 50, 90 μM) for 24 h, exposed to UVB (25 mJ/cm^2^), and then allowed to recover. As expected, after 6 h, UVB alone caused a significant decrease in the mRNA level of *CAT, HO-1,* and *NQO1* compared to the control [47,66,68,69], whereas A02 pre-treatment counteracted this effect (Figure 1F). The Western blot analysis of catalase protein expression at 24 h confirmed the mRNA data (Figure 1G). Furthermore, we analyzed the protein expression level of two DNA damage markers, p53 [70,71] and phosphorylated histone H2A.X (γ-H2AX), the expression of which is associated with the number of double-strand breaks after UV-radiation [72,73]. NHKs pre-treated with A02 showed a reduced level of these UV-related genotoxicity markers compared to those stimulated with UVB alone, suggesting that A02 also exerted a protective effect on UV-induced damage (Figure 1G). The A02 dose of 90 μM proved to be the most effective and hence was selected for the subsequent experiments.

### 3.3. Effects of A02 and Octa on Viability, Proliferation, and Differentiation of A431 Cells

PPARγ activation has been demonstrated to inhibit cell growth and promote differentiation in numerous malignant cell types [37,39,40]. Hence, we investigated the effect on cell viability and the anti-proliferative/pro-differentiative potential of the two compounds on the human cutaneous squamous cell carcinoma (cSCC) A431 cells. The MTT assay indicated no significant effect of Octa and A02 after 72 h of treatment on the NAD(P)H-dependent cellular mitochondrial activity, an indicator of cell viability (Appendix A). Notably, we observed a significant reduction in the number of A431 cells only after A02 treatment (Figure 2A). In compliance with these results, 48 h of treatment with A02 resulted in an increase in cells in the G0/G1 phase of the cell cycle (Figure 2B). To corroborate the A02 effect on the slowing of proliferation, we proceeded to analyze the expression level of the cell cycle regulators p21WAF1/CIP1 (p21) and cyclin D1. A02 significantly induced the expression of the cell cycle progression inhibitor p21 and downregulated the level of the cell proliferation-associated protein cyclin D1 (Figure 2C). Moreover, as for NHKs (see Appendix A), A02 significantly increased both mRNA and protein levels of the differentiation markers filaggrin (FLG), involucrin (IVL), and loricrin (LOR), whereas Octa was not able to modulate their expression (Figure 2D,E, respectively).

### 3.4. A02 and Octa Inhibit TGF-β1-Induced EMT in A431 Cells

EMT plays a crucial role in promoting carcinoma progression and is characterized by cancer cell transition from an epithelial differentiated state to a mesenchymal-like phenotype, more prone to invade the surrounding tissues [23,26,27]. It has been reported that the activation of PPARγ inhibited the induction of EMT markers in lung adenocarcinoma [43,74]. To verify whether Octa and A02 were able to interfere with the EMT process in cutaneous squamous cell carcinoma, A431 cells were pre-treated with the compounds for 1 h and then exposed to the EMT inductor TGF-β1 (15 ng/mL) [42] for 48 h. We evaluated the mRNA expression level of *N-CADHERIN* (*NCAD*), *E-CADHERIN* (*ECAD*), *SLUG*, *FIBRONECTIN, VIMENTIN*, and *MMP-2*, as markers of EMT. Pre-treatment with Octa, and even more with A02, counteracted the TGF-β1-mediated upregulation of *NCAD, FIBRONECTIN, VIMENTIN,* and *MMP2* but was ineffective in preventing the upmodulation of *SLUG* (Figure 3A). The Western blot analysis of vimentin and fibronectin confirmed the ability of Octa and even more of A02 to reduce the increase in protein expression promoted by the stimulation with TGF-β1 for 72 h (Figure 3B). E-cadherin transcript and protein levels were not affected by any of the treatments. In line with these results, qualitative and quantitative analyses of immunofluorescence staining for vimentin and fibronectin demonstrated that both compounds were effective in counteracting the increased expression of the EMT markers induced by TGF-β1 stimulation (Figure 3C). No significant modification of E-cadherin expression was observed following the different treatments. However, the immunofluorescence analysis revealed the delocalization of the protein from the plasma membrane to the cytoplasm associated with a reduced cell–cell adhesion following TGF-β1, which was partially recovered by the treatment with A02 and Octa (Figure 3C). 

Furthermore, we investigated whether Octa and A02 were able to interfere with the increased production of reactive oxygen species (ROS) and the activation of p38 MAPK, AKT, and ERK signaling, known to play an important role in the EMT process [23,26,74]. To this end, A431 cells were pre-treated with Octa and A02 for 1 h and then exposed to TGF-β1. Cells stimulated with the EMT inductor for 24 h showed a significant increase in intracellular ROS levels, and the pre-treatment with both compounds prevented this effect (Figure 4A). In addition, both molecules significantly reduced the levels of p38 MAPK and AKT phosphorylation (Figure 4B) and counteracted the p38 MAPK and ERK activation (Figure 4C) promoted by the treatment with TGF-β1 for 15 min and 24 h, respectively.

Since cell motility favors tumor invasiveness and metastases [75,76], we investigated the effects of both compounds on A431 cell migration promoted by TGF-β1 treatment using the wound scratch assay. The treatment with Octa and A02 significantly counteracted the motogenic effect induced by TGF-β1 as assessed by the measurements of the covered scratched area (Figure 4D).

### 3.5. A02 and Octa Counteract Lipid Metabolic Reprogramming Associated with TGF-β1-Induced EMT in A431 Cells

To confirm the ability of A02 and Octa to counteract the acquisition of invasive and metastatic potential of tumor cells, we evaluated the effects of these compounds concerning the significant changes in lipid metabolism that characterize the EMT process. To this end, A431 squamous carcinoma cells were pre-treated with Octa and A02 for 1 h and then exposed to TGF-β1 (15 ng/mL) for 24 h and 72 h to promote the EMT transition. The lipidomic profile of the A431 cell membranes was acquired, and the monounsaturated/saturated fatty acid ratio (MUFA/SFA), which is positively correlated to the greater aggressiveness of cancer cells [77], was determined. The data obtained demonstrated that Octa and A02 significantly reduce the A431 cell baseline level of the MUFA/SFA ratio after 72 h of treatment (Figure 5A), proving a beneficial effect of the molecules per se against the tumor. Moreover, Octa and A02 pre-treatment was effective in counteracting MUFA accumulation observed after 72 h of TGF-β1 exposure. In particular, A02 reduced the desaturation index even below the A431 basal level, suggesting its higher efficacy compared to Octa in preventing EMT transition (Figure 5A).

Considering that tumor cells can use free FAs (FFAs) as an energy supply by lipolysis for membrane biosynthesis or signaling processes, we analyzed FFA release in the cell medium. The ratio between C16:1/C16:0 and C18:1/C18:0 was reported as an index of SCD-1 activity and MUFA enrichment in the medium. TGF-β1 treatment resulted in a significant release of palmitoleic acid (C16:1) and oleic acid (C18:1), and Octa and A02 effectively counteracted this effect (Figure 5B,C).

The alteration in cholesterol homeostasis is another lipid metabolic reprogramming feature related to tumor initiation and progression. The higher cell cholesterol content associated with the EMT process contributes to lipid raft domains and membrane biophysical property modulation, fueling the maintenance of the mesenchymal state of tumoral cells [28,78]. The determination of cholesterol levels demonstrated that A02 was able to significantly reduce the basal cholesterol content in A431 cells 72 h after the treatment, while Octa did not exert the same effect (Figure 5D). On the other hand, both molecules successfully lowered the TGF-β1-induced increase in cholesterol content. In particular, A02 was more powerful than Octa in significantly counteracting the TGF-β1 effect observed after 24 h and even more after 72 h of treatment (Figure 5D). Otherwise, Octa pre-treatment almost neutralized A431 cholesterol increase, reaching statistical significance only after 72 h (Figure 5D).

Arachidonic acid metabolic pathways, and in particular the generation of bioactive lipids by COX-2 and lipoxygenase (LOX), have been reported to regulate the EMT program in cancer cells [31,79]. Among arachidonic acid metabolites, 5(S)-hydroxyeicosatetraenoic acid (5-HETE), the metabolite of 5-LOX; PGE_2_; PGD_2_; and PGF_2_-α, as representative products of COX-2, were assayed in the cell medium. TGF-β1 significantly increased the extracellular amount of these bioactive lipids, after 24 h and in particular after 72 h of treatment, and Octa and A02 were effective in significantly counteracting this effect (Figure 5E–H).

Altogether, these data demonstrate a key role of the reprogramming of lipid metabolism in the EMT induced by TGF-β1 in A431 cells and suggest that the protective role of Octa and A02 is associated with their ability to preserve the lipid composition of the cell membranes and counteract the release of bioactive lipids involved in the activation of EMT program.

### 3.6. A02 Biological Effects Are Mediated by PPARγ Activation

To investigate whether the biological effects observed in A02-treated A431 could be specifically due to PPARγ activation, we first evaluated p21, phospho-p38 MAPK, and phospho-ERK protein expression in PPARγ-silenced A431 cells. A431 cells were transiently transfected with siRNA for PPARγ (siPPARγ) or control (siCtr), and the real-time RT-PCR and Western blot analyses confirmed that PPARγ expression was efficiently decreased in A431 cells transfected with siPPARγ (Figure 6A). A02 significantly induced the expression of p21 in siCtr-A431, whereas it failed to upregulate the level of this cell cycle regulator in PPARγ-deficient cells (Figure 6B). Moreover, A02 significantly counteracted the induction of p38 MAPK and ERK phosphorylation after TGF-β1 treatment in siCtr cells but failed to prevent their activation in PPARγ-deficient cells (Figure 6C). To evaluate the implication of PPARγ in A02 effects at a longer time, we performed some experiments in the presence of GW9662, a specific and selective PPARγ inhibitor [80]. GW9662 (3 µM) abolished the increase in involucrin and loricrin expression induced by A02 after 72 h of treatment (Figure 6D). Furthermore, A02 failed to counteract the TGF-β1-mediated induction of vimentin and fibronectin in the presence of the inhibitor (Figure 6E). The results document the critical involvement of PPARγ in the biological activity of A02.

### 3.7. A02 Counteracted DMBA-TPA Effects In Vivo

To demonstrate the in vivo relevance of the in vitro results, we induced cutaneous papillomas by DMBA-TPA treatments in the MITO-Luc mouse model. In this transgenic reporter mouse model, luciferase activity is under the control of a cyclin B2 promoter fragment whose activity is strictly dependent on cell proliferation [63,64,65]. Appendix A shows the schematic representation of the experimental design and morphology of papillomas induced in these animals. Since aberrant proliferation represents an early pre-neoplastic event and considering the superiority of A02 in several in vitro endpoints, we investigated the efficacy of this compound in reducing tumor mass in this murine model. Figure 7A shows representative images of a MITO-Luc mouse before induction of papillomas (pre-imaging), at the end of DMBA-TPA treatment (t0), and after 15 treatments (t1) with placebo or A02. As expected, we detected induction of bioluminescence in the treated sites after DMBA-TPA application and, of note, a significant decrease in photon emission after the treatment with the compound. On the contrary, we observed an increase in bioluminescence in placebo-treated lesions (Figure 7B). The treatment with A02 also reduced the number of lesions (Figure 7C) and the tumor area (Figure 7D). Furthermore, we evaluated the mRNA expression levels of the EMT markers NCAD, ECAD, SLUG, FIBRONECTIN, VIMENTIN, α-SMA, and MMP2 in samples of mouse skin collected at the end of the treatments. A02 was able to significantly reduce the expression of all these transcripts in comparison with the placebo-treated mice (Figure 7E).

## 4. Discussion

The rise in the incidence of keratinocyte carcinoma is an important public health concern, and research is continuing to aim at the development of an efficient cSCC treatment that combines higher clearance rates with few side effects, short treatment duration, and low costs. Cutaneous carcinogenesis is primarily mediated by UV exposure and is characterized by DNA damage and alterations of signaling pathways crucially involved in cell proliferation and survival, oxidative stress, inflammation, immunosuppression, differentiation, remodeling, and apoptosis [3,4]. Data in the literature on knockout mouse models suggest a protective role for PPARγ in keratinocyte-derived carcinoma. Mice lacking epidermal PPARγ (Pparg-/-^epi^ mice) show a marked increase in TNF-α expression and photocarcinogenesis associated with UVB-induced apoptosis, inflammation, barrier dysfunction, and epidermal hyperplasia [18,21]. In the same mouse model, the PPARγ agonist rosiglitazone reverses UVB-induced systemic immunosuppression [18]. Moreover, mice lacking epidermal PPARγ or its heterodimerization partner (RXRα), as well as mice with germ-line deletion of one PPARγ allele, are prone to increased chemical carcinogenesis [81,82]. In addition to these “loss of function” mouse models, pharmacological studies suggest that PPARγ activation may exert anti-neoplastic effects through the suppression of tumor-promoting chronic inflammation associated with UV exposure, as well as by strengthening antitumor immune responses in the skin [18,20,83]. Thus, PPARγ agonists may represent a potential pharmacological target in the prevention or treatment of skin cancer. Ligands for PPARγ include a variety of compounds, both natural and synthetic. Most of the natural ligands are fatty acids, fatty acid derivatives, and phytochemicals [5,7]. Thiazolidinediones (TZDs) are synthetic PPARγ ligands currently used in the treatment of type 2 diabetes for their ability to induce insulin sensitization and improve glycemic control [84]. However, TZDs are unlikely candidates for chemoprevention or long-term chemotherapy of cutaneous skin cancer due to their lack of specificity and negative side-effect profile [85,86,87]. In this regard, Octa, a specific PPARγ modulator, represents a particularly attractive candidate for skin photoprotection due to the absence of long-term side effects [48,88]. Moreover, the compound is able to promote multiple beneficial effects against photodamage in the three main skin cell populations via PPARγ activation [11,12,47,51,88]. The present findings support the utility of PPARγ activation by specific modulators, such as A02 and Octa, to prevent and treat keratinocyte-derived skin cancer. This is the first study, to our knowledge, examining the ability of PPARγ activation to induce differentiation markers and to antagonize the TGF-β1-mediated changes associated with the EMT process in a human squamous carcinoma cell line. Our in vitro data with A02 are in agreement with the literature reports on several malignant cell types, suggesting that PPARγ activation inhibits human carcinoma cell growth by regulating the expression of the cell cycle-associated proteins and inducing terminal differentiation [37,38,39,40]. Interestingly, between Octa and A02, only A02 is able to inhibit cell growth and promote pro-differentiative effects. These distinct results may depend on the type of interaction among PPARγ and its ligands that seems to determine the conformational change of the receptor and consequently regulate the activation of different clusters of genes [89]. The docking results indicated an alternative conformation for A02 in terms of PPARγ binding pocket location and ligand orientation compared to the Octa reference. This could explain the different effects we observed for Octa and A02.

The loss of epithelial markers and the acquisition of mesenchymal features are achieved through complex mechanisms involving different signaling pathways, transcription factors, altered expression of adhesion molecules, reorganization of cytoskeletal proteins, and production of extracellular matrix-degrading enzymes [22,23,26,27,90]. The activation of p38 MAPK, AKT, and ERK signaling plays an important role in the EMT process [23,26,74]. In compliance with data obtained for thyroid cancer cells [45] and lung adenocarcinoma cells [74], our results on the A431 squamous cell carcinoma cells treated with Octa or A02 indicate the reversion of the TGF-β1-induced EMT markers and the inhibition of cell migration in association with the downregulation of p38 MAPK, AKT, and ERK phosphorylation.

The analysis of the epithelial phenotype marker E-cadherin in A431 cells after treatment with TGF-β1 did not show a decreased expression of the protein but clearly revealed its delocalization from the plasma membrane to the cytoplasm associated with a reduced cell–cell adhesion. Literature data show a correlation between the membranous E-cadherin expression and the degree of tumor differentiation, with upregulation in well-differentiated cSCC and attenuated or missing staining in poorly differentiated tumors, which are characterized by a high cytoplasmic expression of E-cadherin [91,92]. This translocation from the membrane to the intracytoplasmic region is considered a functional loss of this adhesion molecule, linked to the promotion of the EMT process [93,94].

Reprogramming of lipid metabolism has received increasing recognition as a hallmark of cancer cells because dysregulation of lipids and alteration of related enzyme profiles are closely correlated with oncogenic signals and malignant phenotypes, such as metastasis and EMT [95]. Heightened de novo lipogenesis is required for cellular transformation and cancer progression [96]. Transformed cells and cancerous tissues are characterized by a high amount of MUFAs [97] and the upmodulation of SCD1 [98]. In cancer cells, PPARγ promotes the adaptive regulation of lipid metabolism in response to microenvironment changes, and its modulation by small molecules broadly suppresses EMT progression [99,100]. Our data showed that TGF-β1 induced a significant increase in the MUFA/SFA ratio in A431 cells, indicating the accumulation in cellular membranes of lipids generated by de novo synthesis and overexpression of SCD-1. A02 and Octa significantly counteracted this effect, suggesting that PPARγ modulation interferes with SCD-1 activity and FA metabolism dysregulation associated with the EMT program. Moreover, FA uptake from the microenvironment is altered in cancer cells and contributes to proliferation and dissemination to distant organs [101]. We observed that TGF-β1 caused a significant release of free MUFAs, in particular palmitoleic and oleic acids, in the culture media of A431 cells. This effect suggests the upregulation of lipolytic pathways that mobilize free fatty acids generating oncogenic lipid signals that, in turn, fuel aggressive features of cancer. The treatment with Octa or A02 significantly counteracted the release of free MUFAs, reducing the C16:1/C16:0 and C18:1/C18:0 ratios and limiting the incorporation and the utilization of exogenous MUFAs, thus dampening the pro-oncogenic lipid signaling [102].

Cholesterol dyshomeostasis has emerged as a key requirement for cancer initiation and progression [103]. The increased amount of cholesterol inside cell membranes corresponds to an increase in lipid raft presence, leading to an enhanced response to cell signaling proteins involved in the neoplastic transformation and the activation of the EMT program [78]. PPARγ activation has been reported to reduce cholesterol synthesis [104]. The treatment with TGF-β1 caused the enrichment of cholesterol in cellular membranes of A431 cells, and Octa or A02 reduced significantly this accumulation, suggesting that these molecules might interfere with EMT and also counteract the alteration of lipid raft domains. The tumoral microenvironment is characterized by an inflammatory profile where bioactive lipids, in particular prostanoids and eicosatetraenoic acids, generated by arachidonic acid (AA) metabolic pathways, represent the main drivers of tumor progression and metastasis-initiating processes, such as EMT [105]. 5(S)-Hydroxyeicosatetraenoic acid (5-HETE), the metabolite of 5-lipoxygenase (5-LOX), promotes cancer cell proliferation and can potentially promote EMT by triggering the ERK signaling pathway [106]. The COX-2/PGE2 axis overexpression has been reported in several human cancers, including skin cancer [107], and plays an essential role in activating signaling pathways associated with EMT [108]. In cancer cells, PGD2 caused morphological changes into a mesenchymal-like phenotype and a significant enhancement of invasion and migration, by a mechanism involving the upregulation of TGF-β1 [109]. PGF2α has been shown to promote carcinogen-induced malignant transformation [110] and to stimulate motility and invasion of tumor cells [111]. In our experiments, TGF-β1 induced a sustained release of several AA metabolites, in particular, 5-HETE, PGE2, PGD2, and PGF2α, and the treatment with Octa or A02 significantly reduced this effect, confirming the ability of PPARγ modulators to counteract environmental conditions capable of promoting EMT.

To support the in vitro results in vivo, we took advantage of a murine transgenic model, able to quantify the spatial and temporal progression of carcinogen-induced lesions in the same animal and to identify animal-to-animal variations [53]. This model allowed us to monitor the efficacy of the PPARγ activator A02 in reducing keratinocyte-derived carcinoma progression over time. Our results highlighted the effectiveness of the molecule in reducing the proliferation of induced lesions and the expression of typical EMT markers. These findings are in agreement with the previously reported ability of a PPARγ activation to significantly promote a more epithelial, and less mesenchymal, phenotype in a murine lung squamous cell carcinoma model [44] and to exert anti-neoplastic effects through inhibition of cell growth and promotion of differentiation [37,39,40].

In conclusion, in the present work, we propose the new compound A02 as a PPARγ modulator capable of counteracting the appearance of AK and its progression to SCC. Since AK represents the first sign of severe chronic ultraviolet radiation exposure, the ability of the compound to stimulate cellular antioxidant defense and protection from UV-induced DNA damage in NHKs might prevent AK development. Furthermore, A02 counteracts the TGF-β1-induced EMT in cSCC cells and in the murine tumor model, suggesting its efficacy in avoiding AK-to-cSCC progression and promoting the remission of existing AK lesions. Taken together, the present findings provide a rationale for further preclinical/clinical studies to develop new therapeutic approaches for the prevention and/or treatment of keratinocyte-derived cancer through the activation of PPARγ. Hence, a major challenge for future research is to dissect how the different patterns of genes activated by PPARγ and involved in protective effects against UV-induced inflammation, oxidative stress, DNA damage, and photocarcinogenesis can be targeted by conceiving more selective PPARγ ligands with a low toxicity profile, thus also reducing undesirable side effects. A great deal of work is necessary to fully characterize the pharmacological activity and specificity of these agents.

## Figures and Tables

**Figure 1 cells-12-01007-f001:**
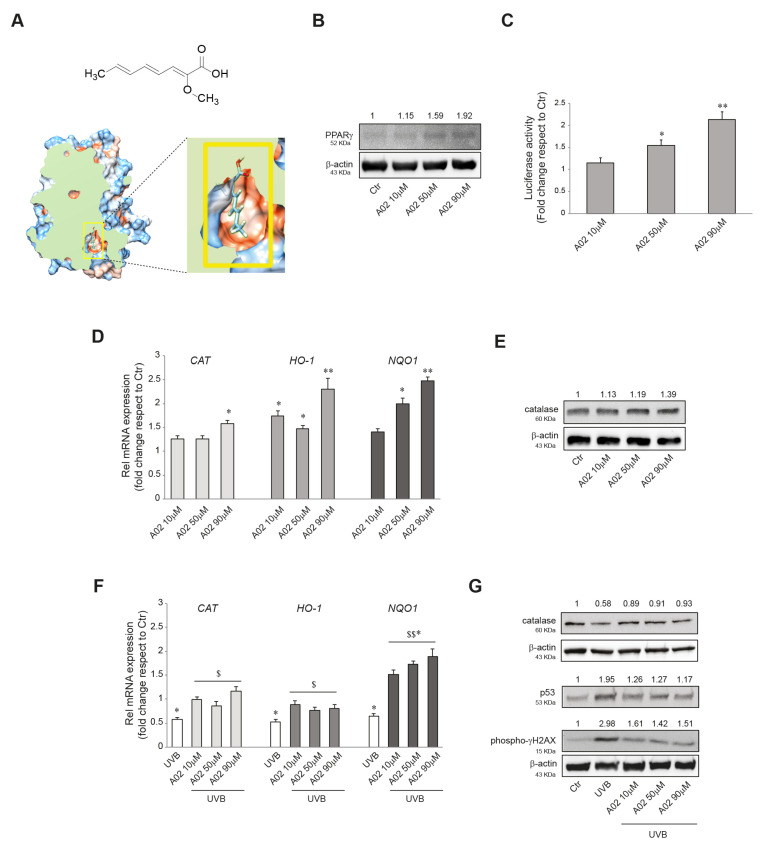
A02 induces PPARγ expression and activation, promotes antioxidant defense, and protects from UVB-induced damages in NHKs. (**A**) Schematic representation of the chemical structure of A02 and its 3D molecular model of docking to PPARγ human receptor. (**B**) Western blot analysis of PPARγ protein expression in NHKs treated with A02 (10–50–90 µM) for 24 h. Representative blots are shown. (**C**) Luciferase activity analysis of NHKs transfected with pGL3-(Jwt)3TKLuc reporter construct. After 24 h of transfection, cells were treated with A02 (10–50–90 µM) for 24 h. The variability of transfection was normalized with Renilla luciferase activity. The data are presented as the mean ± SD of three independent experiments and are expressed as the fold change with respect to untreated control cells (* *p* < 0.05, ** *p* < 0.01 vs. untreated control). (**D**) Real-time RT-PCR analysis of *CAT*, *HO-1,* and *NQO1* in NHKs treated with A02 (10–50–90 µM) for 6 h. All mRNA values were normalized against the expression of *GAPDH* and were expressed relative to untreated control cells. The data in the graphs are the mean ± SD of three independent experiments (* *p* < 0.05, ** *p* < 0.01 vs. untreated control). (**E**) Western blot analysis of catalase protein expression in NHKs treated with A02 (10–50–90 µM) for 24 h. Representative blots are shown. (**F**) Real-time RT-PCR analysis of *CAT*, *HO-1,* and *NQO1* in NHKs pre-incubated with A02 (10–50–90 µM) for 24 h and then irradiated with UVB 25 mJ/cm^2^. mRNA was extracted 6 h after the irradiation. All mRNA values were normalized against the expression of *GAPDH* and were expressed relative to untreated control cells. The data in the graphs are mean ± SD of three independent experiments (* *p* < 0.05 vs. untreated control; ^$^ *p* < 0.05, ^$$^ *p* < 0.01 vs. UVB-treated cells). (**G**) Western blot analysis of catalase, p53, and phospho-γH2AX protein expression in NHKs pre-incubated with A02 (10–50–90 µM) for 24 h and then irradiated with UVB 25 mJ/cm^2^. Proteins were extracted 24 h after the irradiation. Representative blots are shown. β-actin was used as an endogenous loading control for Western blot analysis. Densitometric scanning of band intensities was performed to quantify the change in protein expression. Data represent the mean ± SD of three independent experiments and are expressed as fold change with respect to untreated control cells (control value taken as 1-fold in each case).

**Figure 2 cells-12-01007-f002:**
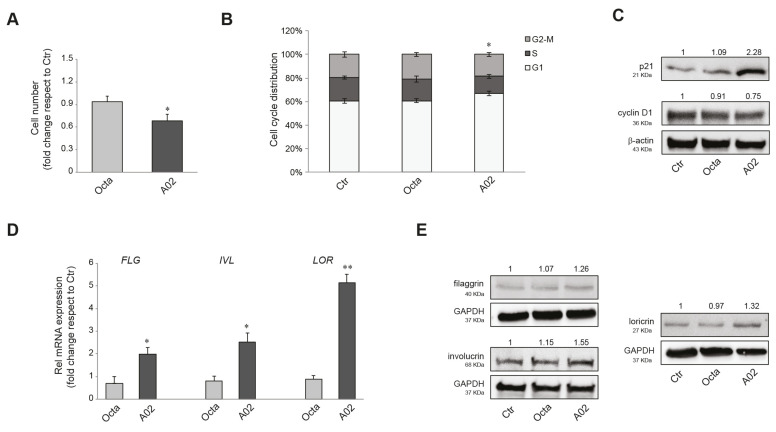
Effects of A02 and Octa on viability, proliferation, and differentiation of A431 cells. (**A**) Cell number analysis was performed in A431 treated with Octa or A02 (90 µM) for 72 h. The data are presented as the mean ± SD of three independent experiments and are expressed as the fold change with respect to untreated control cells (* *p* < 0.05 vs. untreated control). (**B**) Cell cycle distribution evaluated by flow cytometric analysis on A431 treated with Octa or A02 (90 µM) for 48 h. The bar graph shows the distribution of cells among the different phases of the cell cycle. The data are expressed as the mean ± SD of three independent experiments (* *p* < 0.05 vs. untreated control). (**C**) Western blot analysis of p21 and cyclin D1 protein expression in A431 treated with Octa or A02 (90 µM) for 24 h and 48 h, respectively. Representative blots are shown. (**D**) Real-time RT-PCR analysis of *FLG, IVL,* and *LOR* in A431 treated with Octa or A02 (90 µM) for 48 h. All mRNA values were normalized against the expression of *GAPDH* and were expressed relative to untreated control cells. The data in the graphs are mean ± SD of three independent experiments (* *p* < 0.05, ** *p* < 0.01 vs. untreated control). (**E**) Western blot analysis of filaggrin, involucrin, and loricrin protein expression in A431 treated with Octa or A02 (90 µM) for 72 h. β-actin and GAPDH were used as endogenous loading control for Western blot analyses. Densitometric scanning of band intensities was performed to quantify the change in protein expression. Data represent the mean ± SD of three independent experiments and are expressed as fold change with respect to untreated control cells (control value taken as 1-fold in each case).

**Figure 3 cells-12-01007-f003:**
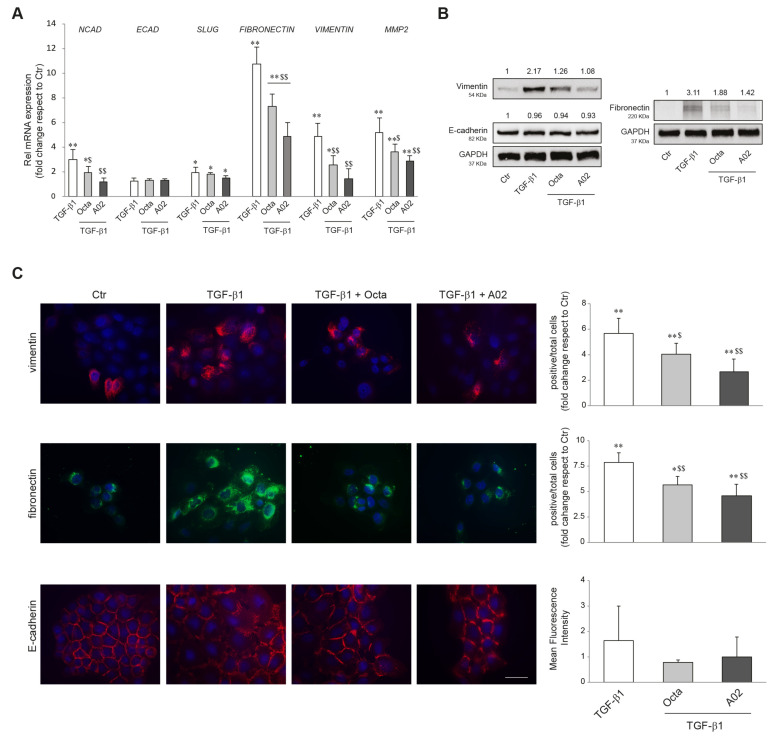
Effects of A02 and Octa on the TGF-β1-induced EMT in A431 cells. (**A**) Real-time RT-PCR analysis of *NCAD, ECAD, SLUG, FIBRONECTIN, VIMENTIN,* and *MMP2* in A431 pre-incubated with Octa or A02 (90 µM) for 1 h and then treated with TGF-β1 (15 ng/mL). All mRNA values were normalized against the expression of *GAPDH* and were expressed relative to untreated control cells. The data in the graphs are the mean ± SD of three independent experiments (* *p* < 0.05, ** *p* < 0.01 vs. untreated control; ^$^ *p* < 0.05, ^$$^ *p* < 0.01 vs. TGF-β1-treated cells). (**B**) Western blot analysis of vimentin, E-cadherin, and fibronectin protein expression in A431 pre-incubated with Octa or A02 (90 µM) for 1 h and then treated with TGF-β1 (15 ng/mL) for 72 h. GAPDH was used as an endogenous loading control for Western blot analysis. Densitometric scanning of band intensities was performed to quantify the change in protein expression. Data represent the mean ± SD of three independent experiments and are expressed as fold change with respect to untreated control cells (control value taken as 1-fold in each case). (**C**) Immunofluorescence and quantitative analysis of vimentin, fibronectin, and E-cadherin in A431 cells. Results are expressed as fold change of positive cells or mean fluorescence intensity with respect to control. Nuclei were counterstained with DAPI. Bar: 20 μm. (* *p* < 0.05, ** *p* < 0.01 vs. control; ^$^ *p* < 0.05 vs. TGF-β1; ^$$^ *p* < 0.01 vs. TGF-β1).

**Figure 4 cells-12-01007-f004:**
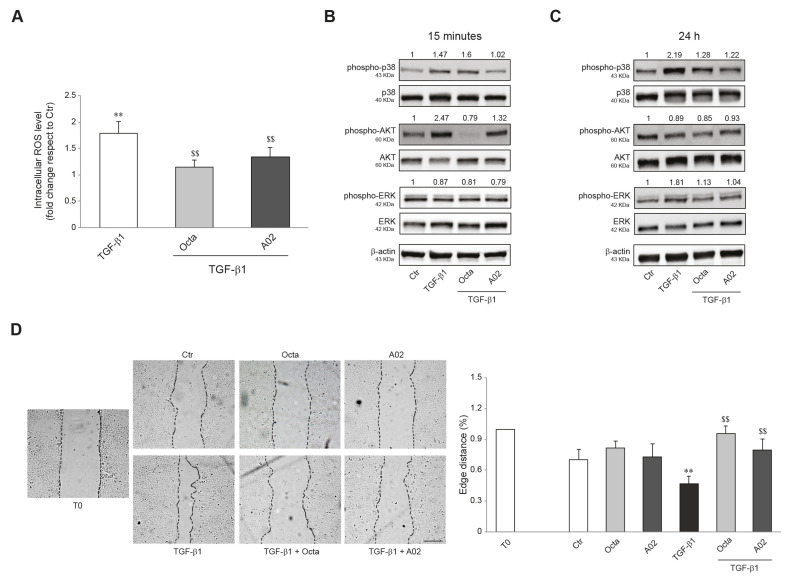
Effects of A02 and Octa on ROS generation, inflammation, and migration in A431 cells. (**A**) ROS levels in A431 pre-treated with Octa and A02 (90 µM) for 1 h and then exposed to TGF-β1 (15 ng/mL) for 24 h. Results are expressed as the fold change with respect to untreated control cells (** *p* < 0.01 vs. untreated control; ^$$^ *p* < 0.01 vs. TGF-β1-treated cells). Western blot analysis of phospho-p38, p38, phospho-AKT, AKT, phospho-ERK, and ERK protein expression in A431 pre-incubated with Octa or A02 (90 µM) for 1 h and then treated with TGF-β1 (15 ng/mL) for 15 min (**B**) and 24 h (**C**). β-actin was used as an endogenous loading control for Western blot analyses. Densitometric scanning of band intensities was performed to quantify the change in protein expression. Data represent the mean ± SD of three independent experiments and are expressed as fold change with respect to untreated control cells (control value taken as 1-fold in each case). (**D**) Representative images of the scratch assay of A431 cells treated with Octa or A02 alone or after TGF-β1 stimulation at T0 and 24 h. Bar: 200 μm. Quantitative analysis resulting from the measurements of the covered scratched area. The results are expressed as fold change with respect to T0 which was set as 1 by definition (** *p* < 0.01 vs. T0; ^$$^ *p* < 0.01 vs. TGF-β1).

**Figure 5 cells-12-01007-f005:**
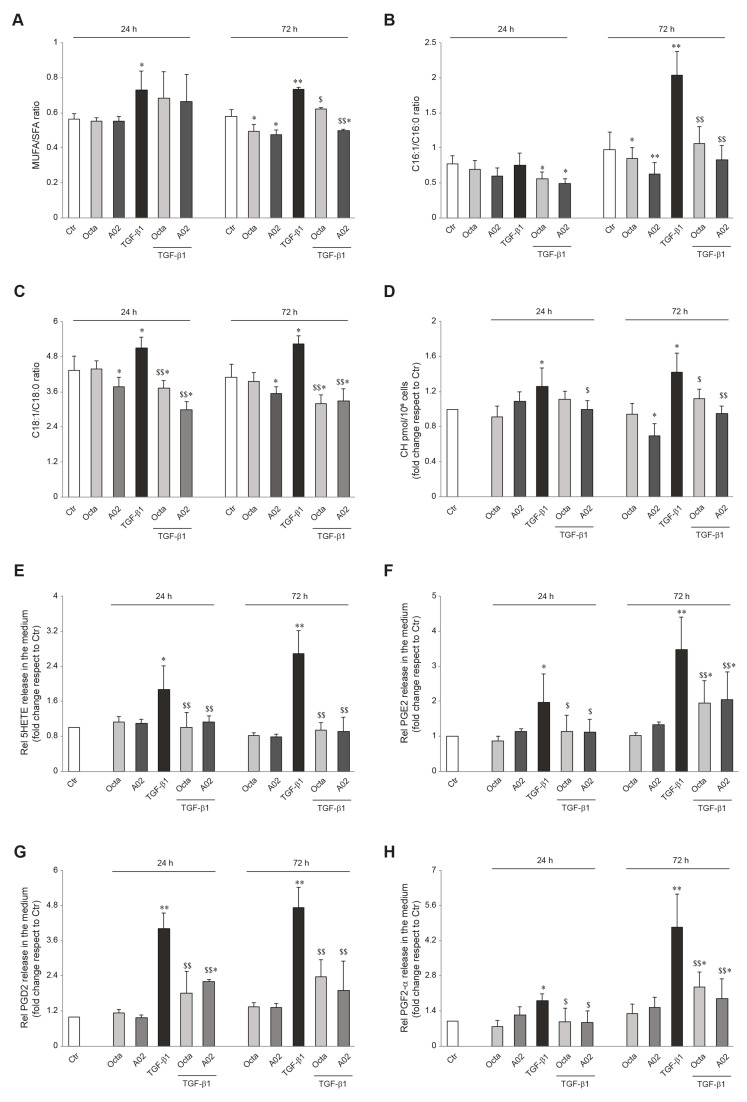
A02 and Octa effects against the alteration of cellular and extracellular lipid composition following the TGF-β1-induced EMT in A431 cells. Quantification by GCMS of (**A**) cell membrane MUFA/SFA ratio, (**B**) extracellular C16:1/C16:0 ratio, (**C**) extracellular C18:1/C18:0 ratio, and (**D**) cholesterol amount in A431 cells pre-treated with Octa and A02 (90 µM) for 1 h and then exposed to TGF-β1 (15 ng/mL) alone or in combination with Octa and A02 for 24 h and 72 h. Quantification by LC-MS/MS of (**E**) 5HETE, (**F**) PGE2, (**G**) PGD2, and (**H**) PGF2 released in the medium by A431 cells pre-treated with Octa and A02 (90 µM) for 1 h and then exposed to TGF-β1 (15 ng/mL) alone or in combination with Octa and A02 for 24 h and 72 h. The results are expressed as a ratio or as fold change with respect to control. (* *p* < 0.05 vs. control; ** *p* < 0.01 vs. control; ^$^ *p* < 0.05 vs. TGF-β1; ^$$^ *p* < 0.01 vs. TGF-β1).

**Figure 6 cells-12-01007-f006:**
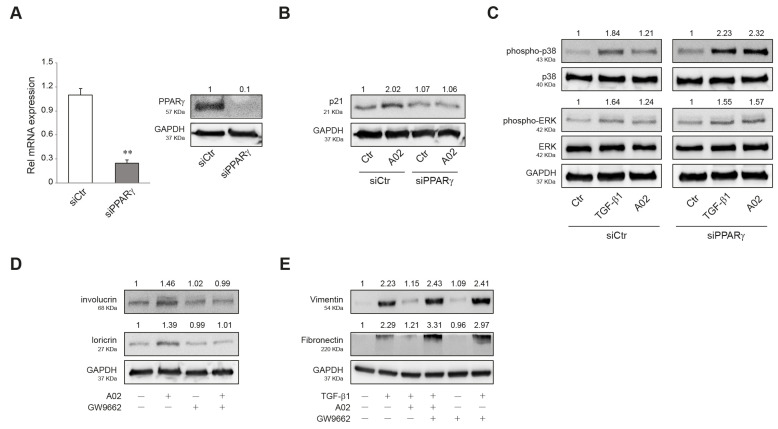
PPARγ activation is needed to promote A02 biological effects. (**A**) A431 cells were transfected with siRNA specific for PPARγ (siPPARγ) or non-specific siRNA (siCtr). PPARγ level was evaluated by real-time RT-PCR and Western blot analysis 24 h after transfection. The mRNA values were normalized against the expression of *GAPDH*. The data in the graphs are the mean ± SD of three independent experiments (** *p* < 0.01 vs. siCtr cells). Representative blots are shown. Western blot analysis of (**B**) p21 and (**C**) phospho-p38, p38, phospho-ERK, and ERK protein expression in A431 cells transfected with siPPARγ and siCtr exposed to A02 (90 µM) for 1 h and then treated with TGF-β1 (15 ng/mL) for 24 h. Representative blots are shown. Western blot analysis of (**D**) involucrin and loricrin and (**E**) vimentin and fibronectin protein expression in A431 pre-incubated with GW9662 (3 µM) for 1 h, exposed to A02 (90 µM) for 1 h, and finally treated with TGF-β1 (15 ng/mL) for 72 h. GAPDH was used as an endogenous loading control for Western blot analysis. Densitometric scanning of band intensities was performed to quantify the change in protein expression. Data represent the mean ± SD of three independent experiments and are expressed as fold change with respect to siCtr cells (**B**,**C**) and untreated control cells (**D**,**E**) (control value taken as 1-fold in each case).

**Figure 7 cells-12-01007-f007:**
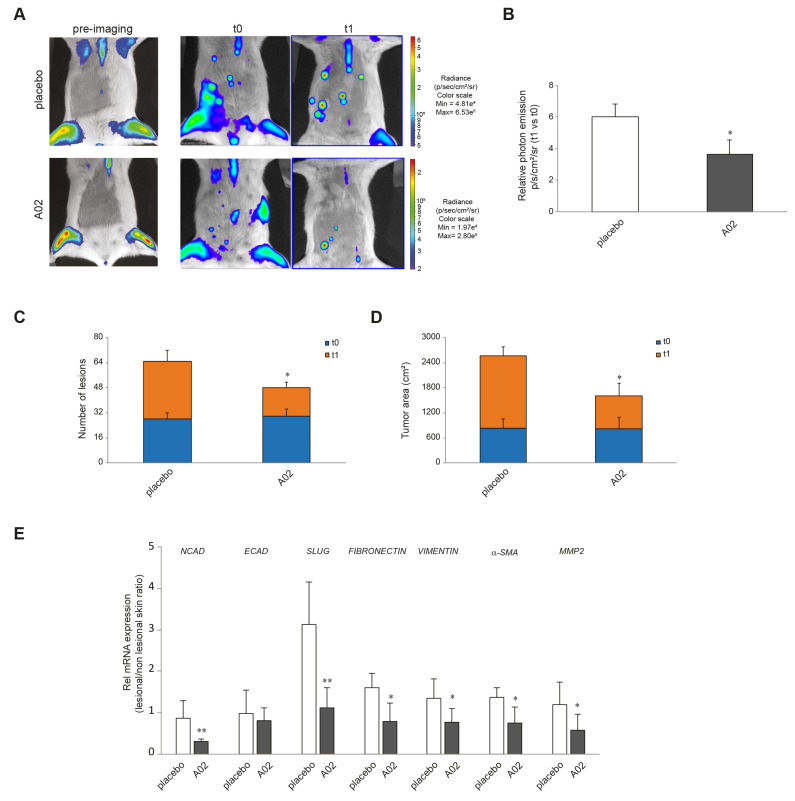
A02 impact on DMBA-TPA effects in vivo. (**A**) Luciferase activity in MITO-Luc mouse with induced papillomas. BLI of MITO-Luc mice before treatment (pre-imaging), after treatment with DMBA-TPA on the ventral abdomen (t0), and after 15 treatments with placebo or A02 (t1). Images were collected for 5 animals for each treatment, and a representative animal is shown. Photon emission from the lesions was measured as photons per second per square centimeter per steradian (photons/s/cm^2^/sr). (**B**) The graph illustrates BLI signal intensities relative to DMBA-TPA drug treatment control (t0) after 15 treatments with placebo or A02 (t1). Each bar represents the mean value ± SEM of five animals. The number of lesions (**C**) and tumor area (cm^2^) (**D**) have been measured at t0 and t1 by automatically drawing the regions of interest upon BLI analysis. The graph represents the mean value ± SEM of five animals. (**E**) Real-time RT-PCR analysis of NCAD, ECAD, SLUG, FIBRONECTIN, VIMENTIN, α-SMA, and MMP2 in samples of mouse skin collected in lesional and nonlesional areas at the end of the 15 treatments with placebo or A02 (3 different mice per treatment group). All mRNA values were normalized against the expression of GAPDH and were expressed relative to nonlesional skin (* *p* < 0.05, ** *p* < 0.01 vs. placebo).

## Data Availability

The datasets used and/or analyzed during the current study are available from the corresponding author upon reasonable request.

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
