# Peer review of "The Activation of PPARγ by (2Z,4E,6E)-2-methoxyocta-2,4,6-trienoic Acid Counteracts the Epithelial–Mesenchymal Transition Process in Skin Carcinogenesis"

_cells, 2023, doi:10.3390/cells12071007_

Round 1

Reviewer 1 Report

In this manuscript Flori et al. study the effects of PPARg agonist on cutaneous squamous cell carcinoma (SCC) in vitro and in vivo, using a chemical carcinogenesis protocol. This is an interesting and well-written work with a rather comprehensive experimental approach to determine the effect of the A02 compound on SCC.

There are however few concerns of this reviewer that should be addressed.

1) Flori and co-workers associate the various biological effects obtained with A02 to PPARg, however they do not present any evidence that this is the case. It would be expected that if PPARg is required for A02 beneficial effects, these effects should be abolished in the absence of PPARg. Thus, PPARg knockdown or knockout should be used to establish a causative link between A02 and PPARg.

2) To confirm and expand the observation obtained in vivo it would be useful to indicate whether the size, number, or histology of the papillomas in the DMBA/TPA experiment is significantly different between A02-treated and untreated mice.

Minor comments

1) Since (2Z,4E,6E)-2-methoxyocta-2,4,6-trienoic acid (A02) is a novel compound that apparently has not been characterized previously, the chemical structure should be more clearly indicated in Fig.1A.

2) In the Figure, a point instead than a comma should be used for decimal separators.

Reviewer 2 Report

Dear Authors,

thank you very much for preparing this important manuscript which is focused on skin cancerogenesis. As the number of skin cancer cases rapidly increases it is reasonable to search for compounds that can inhibit the development of cancer disease. In this manuscript, the Authors have investigated the possible role of the PPARgamma activators (Octa and A02) on a very complex model (in silico docking, in vitro investigations on the primary keratinocytes and commercial A431 cell line, and in vivo model on DMBA-treated mice). The study was well-designed and the manuscript is clearly written.

However, I have some suggestions for improving this manuscript:

1. please italicize all words like in silico (line 108), ad libitum (line 368), via (line 675), de novo (line 713)

2. please unify the writing of values and number in the whole manuscript: with space (2 mM, 320 nm, 10 cm, 1 ml, 15 ng/ml, 72 h) and without space (37°C, -80°C)

3. in subsection 2.18, please change "mL" and "µL" to "ml" and "µl"

4. line 345, please change the reference from (Furugen et al., 2015; Le Faouder et al., 2013) to the corresponding numbers

5. line 192, please change "UVB lamps do have not UVC emission" to "UVB lamps do not have UVC emission"

6. in subsection 2.4, please double-check the applied concentration of the penicillin/streptomycin solution as there is no 100 μg/mL concentration available on the market. Usually, this solution contains 10,000 units/ml of penicillin and 10,000 µg/ml of streptomycin and we dilute it 100 times in the medium.

7. In terms of "Quantitative real-time RT-PCR", although many people use this name, in my opinion, it is a mistake. I would select one of the terms: real-time RT-PCR, quantitative RT-PCR, RT-qPCR and use it uniformly throughout the manuscript.

8. when writing about the mRNA values normalization - GAPDH should be italicized (lines: 420, 425, 482, 531)

9. in the description of figure 3, line 530, please remove NHKs.

10. it is not clear why the Authors have used for western blot two loading controls (beta-actin and GAPDH) and for the qPCR only one (GAPDH). Actually, taking into consideration the introduction of this manuscript (metabolic reprogramming in cancer cells, lines 58-63) and the generally known changes in the cancer cell cytoskeleton (in particular in the actin filaments) both of the applied controls can significantly falsify the results. It would be recommended to keep one control in the whole manuscript or to present results for both controls in a single experiment that do not change these result.

With kind regards

Round 2

Reviewer 1 Report

The authors have accurately responded to all my requests